# Disabling VEGF-Response of Purkinje Cells by Downregulation of *KDR* via miRNA-204-5p

**DOI:** 10.3390/ijms22042173

**Published:** 2021-02-22

**Authors:** Julian Gehmeyr, Abdelouahid Maghnouj, Jonas Tjaden, Matthias Vorgerd, Stephan Hahn, Veronika Matschke, Verena Theis, Carsten Theiss

**Affiliations:** 1Department of Cytology, Institute of Anatomy, Ruhr-University Bochum, Universitaetsstr. 150, Building MA, Level 5, 44780 Bochum, Germany; julgeh@t-online.de (J.G.); Jonas.Tjaden@ruhr-uni-bochum.de (J.T.); veronika.matschke@rub.de (V.M.); Verena.theis@rub.de (V.T.); 2Clinical Research Centre (ZKF), Department of Molecular Gastrointestinal Oncology, Ruhr-University Bochum, Universitaetsstr. 150, 44801 Bochum, Germany; abdelouahid.maghnouj@rub.de (A.M.); stephan.hahn@rub.de (S.H.); 3Neuromuscular Center Ruhrgebiet, Department of Neurology, University Hospital Bergmannsheil, Ruhr-University Bochum, Buerkle-de-la-Camp-Platz 1, 44789 Bochum, Germany; matthias.vorgerd@bergmannsheil.de

**Keywords:** VEGF, VEGFR-2, KDR, miRNA-204-5p, Purkinje cells, cerebellum, organotypic slice cultures, microinjection, dendritic growth

## Abstract

The vascular endothelial growth factor (VEGF) is well known for its wide-ranging functions, not only in the vascular system, but also in the central (CNS) and peripheral nervous system (PNS). To study the role of VEGF in neuronal protection, growth and maturation processes have recently attracted much interest. These effects are mainly mediated by VEGF receptor 2 (VEGFR-2). Current studies have shown the age-dependent expression of VEGFR-2 in Purkinje cells (PC), promoting dendritogenesis in neonatal, but not in mature stages. We hypothesize that microRNAs (miRNA/miR) might be involved in the regulation of VEGFR-2 expression during the development of PC. In preliminary studies, we performed a miRNA profiling and identified miR204-5p as a potential regulator of VEGFR-2 expression. In the recent study, organotypic slice cultures of rat cerebella (postnatal day (p) 1 and 9) were cultivated and VEGFR-2 expression in PC was verified via immunohistochemistry. Additionally, PC at age p9 and p30 were isolated from cryosections by laser microdissection (LMD) to analyse VEGFR-2 expression by quantitative RT-PCR. To investigate the influence of miR204-5p on VEGFR-2 levels in PC, synthetic constructs including short hairpin (sh)-miR204-5p cassettes (miRNA-mimics), were microinjected into PC. The effects were analysed by confocal laser scanning microscopy (CLSM) and morphometric analysis. For the first time, we could show that miR204-5p has a negative effect on VEGF sensitivity in juvenile PC, resulting in a significant decrease of dendritic growth compared to untreated juvenile PC. In mature PC, the overexpression of miR204-5p leads to a shrinkage of dendrites despite VEGF treatment. The results of this study illustrate, for the first time, which miR204-5p expression has the potential to play a key role in cerebellar development by inhibiting VEGFR-2 expression in PC.

## 1. Introduction

The vascular endothelial growth factor (VEGF) was first described by Senger et al. [1] as a tumour-secreted vascular permeability factor (VPF). VEGF-A, in the following referred to as VEGF, was the first member of a family belonging to homodimeric disulphide bound glycoproteins. VEGF-B, -C, -D, -E, -F, and the placenta growth factor (PIGF) also belong to this family [2,3]. At least nine different isoforms of VEGF have been identified so far, all generated as a result of alternate mRNA splicing [4]. Among these isoforms, VEGF-A_165_ is the strongest binding partner of its receptor and, therefore, the biologically most active form of VEGF in mammalians [5]. Best known for its role in angiogenesis and vasculogenesis [6], VEGF is especially expressed under hypoxic, ischemic, or traumatic conditions [7] to promote vascular development, permeability, and endothelial outspreading [6,8]. This growth factor has gained clinical relevance due to its tumour promoting effects, which make it a target of tumour therapeutics such as the monoclonal antibody bevacizumab [9]. This is also used against age-related macular degeneration due to subretinal neovascularisation [10].

VEGF unfolds its effects via the receptor tyrosine kinases VEGFR-1 (*FLT1*), -2 (*KDR, FLK1*), and -3 (*FLT4*)—transmembrane proteins containing a tyrosine kinase sequence in their intracellular part. The VEGF binding site, built by seven immunoglobulin-like domains, is located at the extracellular part of the receptor [11,12]. Tyrosine kinase inhibitors, such as axitinib, can perturb VEGF pathways by blocking the autophosphorylation of VEGF receptors [13], and are used in clinical studies against various tumour diseases [14,15].

Through VEGFR-2, the receptor that conveys most of the VEGF-driven processes, VEGF mediates endothelial migration, and survival, vasculogenesis, and angiogenesis. VEGFR-2 is also strongly expressed in neuronal cells and plays a key role in proliferation, migration and survival as well as axonal guidance and dendritogenesis of neuronal cells [2,16,17,18]. Although VEGF has higher affinity to VEGFR-1 than to VEGFR-2, most of the various effects promoted by VEGFR-1, e.g., angiogenesis, haematopoiesis, neuronal survival, and inflammatory cell recruitment, are triggered by the ligand PIGF. Regarding VEGF, VEGFR-1—especially the soluble VEGFR-1 (sVEGFR-1)—plays more of an inhibitory role as it binds VEGF strongly without triggering any measurable effects [19,20]. 

There has been much interest recently in the role of VEGF as a neurogenic, neurotrophic and neuroprotective factor in the nervous system [21,22]. Besides its effects on neuronal proliferation and migration [23], VEGF provides neuronal regeneration, neurite outgrowth, axonal, and neuronal growth guidance as well as neuronal cell survival [19,24,25,26,27]. The impact of VEGF as a stimulator of neurogenesis during development, maturation and regeneration processes of neuronal tissue especially has been in the focus of current investigations [11]. Recent studies pointed out that VEGF is a key player for dendritogenesis, e.g., of Purkinje cells (PC) in the cerebellum [16] and interneurons in the olfactory bulb [28]. In the adult nervous system, VEGF stimulates neuronal recovery from injuries after incidents, such as strokes or status epilepticus [29,30,31], while low levels of VEGF negatively influence the survival of PC [32] and neuronal injury outcome [33]. Although we know that the expression of VEGF is induced by hypoxia inducible factor 1α (HIF-1α) [7] and most of the VEGF triggered processes on the nervous system are controlled via the MAPK pathway of VEGFR-2 [18], little is known to date about the regulation of VEGF impact on neuronal cells.

Since microRNAs (miRNAs/miR) intervene at numerous points of cell metabolism, such as proliferation, differentiation, and apoptosis by inducing posttranscriptional gene silencing [34,35], they are particularly suitable as modulators of the VEGF pathway [36]. MiRNAs are small, approximately 17–25 nucleotides long, single-stranded, non-coding RNA sections that are able to bind complementary at the 3′UTR region of mature messenger RNA (mRNA) by specific base pairing. This allows them to regulate translational gene expression via RNA interference (RNAi) [37]. If there is a strong complementarity between a miRNA and its associated target mRNA, mRNA degradation is initiated resulting in a decreased expression of the target protein [38,39]. 

Within recent decades, several miRNAs could be identified to participate in the regulation of neurogenesis and the VEGF pathway. While Schaefer et al. [40] found that the complete absence of miRNAs leads to neurodegeneration in rat PC, Zhao et al. were able to prove an upregulation of VEGF by miR-204 in tilapia under hypoxic conditions [36]. By means of a miRNA profiling in microdissected PC, different miRNAs have been identified, which are distributed differently during the development of the cerebellum. These are, therefore, most likely to be involved in PC maturation processes [41]. For one of the identified miRNAs, miR204-5p, which can bind *KDR* [42], many interactions with VEGF-confined processes are already known. Its inhibition of corneal neovascularisation [43], vasculogenic mimicry in breast cancer cells [44] and induction of apoptosis in dopaminergic cells [45] make miR204-5p a promising subject for further investigations of the regulation of VEGF triggered neuronal maturation. We directly investigated the effect of miR204-5p on the VEGF receptor expression in PC of organotypic cerebellar cultures, since it is possible to imitate the effect of miRNAs on cells by means of artificial RNA sequences (miRNA mimics) [46]. Due to the fact that VEGF seems to be crucially involved in the physiological maturation process of PC, mainly at early developmental stages [16], for clinical reasons the possible reactivation of VEGF effects in adult PC has become a point of focus. For this reason, our aim of the current study was to unveil the effect of miR204-5p on VEGF receptor expression and dendritogenesis in neonatal, juvenile, and adult PC. By using immunohistochemistry (IHC) and laser microdissection (LMD) followed by quantitative reverse transcription polymerase chain reaction (qRT-PCR), we conducted research on the differential expression of the VEGF receptors. To gain a deeper insight into the influences of miRNAs on the VEGF triggered development of cerebellar neurons our study was performed with organotypic rat cerebellar slice cultures. Single PC (juvenile and mature) were microinjected with a plasmid to bring in miR204-5p and discover its effects on the VEGFR-2 pathway by morphometric analysis after VEGF treatment. For that, the huge dendritic arbour of mature PC in rat cerebella, growing within the first 4 weeks after birth, offered unique conditions for morphometric analysis of structural changes during the maturation process [47].

## 2. Results

### 2.1. Morphology of PC in Slice Cultures and Cryosections at p9, p15, and p30

In our study, the investigation and morphometric analysis of neonatal, juvenile, and mature PC in rat cerebellar roller tube slice cultures was a key trial to observe their response to VEGF depending on miR204-5p microinjection. Although PC functionality and structure in organotypic slices has already been proven in previous studies [16,48,49], comparability between PC development in organotypic slices and in vivo had to be ensured. For this purpose, we first tested whether the characteristic organisation of the cerebellum was still retained in cerebellar slice cultures. Using IHC and confocal laser scanning microscopy (CLSM), we compared organotypic cerebellar slices of p1 and p9 pups after different ex vivo incubation times (p1 + 8div, p1 + 14div and p9 + 21div) with cerebellar cryosections of rats at p9, p15, and p30 (Appendix A). 

In the course of its development, the cerebellar cortex, still immature at birth, goes through several characteristic changes in structure, which we were able to display both in cryosections (Appendix A) and organotypic cultures (Appendix A). Within the first days after birth (Appendix A), the cerebellum shows a four-layered cortex, consisting of a molecular cell layer (MCL), a line-shaped arrangement of PC in the PC layer (PCL), granule cells forming the granular cell layer (GCL) and a separate external GCL (eGCL). Moreover, in organotypic slices at p1 + 8div a characteristic arrangement including the eGCL was clearly visible (Appendix A). In matured cerebella, the eGCL completely disappeared and the cells are finally arranged in a three-layered cortex in p30 cryosections as well as slice cultures at p9 + 21div (Appendix A). In addition, the PC developmental processes could be made visible in both in vivo and in vitro (Appendix A). PC dendrites developed from a non-polar perisomatic order (Appendix A) to a heavily branched dendritic arbour containing numerous dendritic spines (Appendix A).

This shows that, although the natural arrangement of cells is slightly loosened in organotypic slice cultures, characteristic cerebellar structures are nevertheless clearly visible. PC develop an age-appropriate dendritic tree and keep their organisation in relation to other cells also in cerebellar roller-tube slices, which therefore represent functional neuronal tissue and are highly suitable for studying PC development. As the slice cultures won from neonatal rat pups (≤p10) can be cultivated for up to three weeks without loss of the organotypic cerebellar morphology, this setting is suitable for long-term incubation studies to analyse the effects of miR204-5p on PC and to gain deeper insights into the mechanisms of VEGF-triggered PC development at different maturation levels.

### 2.2. VEGF-Receptor Expression in the Course of PC Development

To determine any impact of miR204-5p on VEGF dependent PC development at different cerebellar maturation stages, the expression levels of all VEGF-receptors were analysed in neonatal, juvenile, and mature PC (p9, p15, and p30) by applying laser microdissection and qRT-PCR (Figure 1). Moreover, we compared the protein level expression of VEGFR-2, which is likely to mediate VEGF effects in neonatal, juvenile and mature PC in vivo as well as in organotypic slice cultures (p1 + 8div, p1 + 14div, p9 + 21div) via IHC (Figure 2 and Figure 3). 

In order to analyse age-dependent mRNA expression levels of *KDR* (coding for VEGFR-2), *FLT1* (coding for VEGFR-1), *FLT4* (coding for VEGFR-3) and Neuropilin-1 and -2 (*NRP1/NRP2*), we processed LMD of single PC from cryoslices of p9 and p30 cerebella after methylene blue staining (Figure 1a). Gene expression values of qRT-PCR are given in relation to *KDR* expression at p9. Our results indicate that *KDR* is by far the most abundant receptor mRNA in PC (Figure 1b). All other gene expression values kept below a ratio of 10% compared to *KDR* gene expression at p9, while the amount of *FLT4* gene product was undetectable in adult development stages. While *KDR* expression level decreased down to 45% at p30 (Figure 1b) and the expression of *NRP1* and *FLT1* also declined during PC development, no significant changes of *NRP2* could be recognised between the maturation levels (Figure 1b). 

IHC studies show a similar result concerning VEGFR-2 expression in PC. While an intense signal of VEGFR-2 was detectable in neonatal PC soma and dendrites (Figure 2a,d,g), positive VEGFR-2 labelling slightly degraded in dendrites and soma of juvenile PC (Figure 2b,e,h). In mature rat cerebella, the intensity of VEGFR-2 signals decreased, and only scattered PC soma and dendrites continued to show a weak signal (Figure 2e,f,i). Incidentally, a positive VEGFR-2 signal could also be detected in fibres of adjacent neurons in the MCL of neonatal cryoslices (Figure 2d,g). Based on the typical localisation and morphology, these are supposed to be Bergmann glia cells. 

VEGFR-2 expression in PC in organotypic slice cultures was similar to in vivo VEGFR-2 expression at all maturation levels (Figure 3), showing a decrease of signal intensity during development. Moreover, filamentous VEGFR-2 signals, such as Bergmann glia signals in cryosections could be detected in neonatal slice cultures (Figure 3d,g), but they cannot be clearly assigned to Bergmann glia cells in that case, as the cerebellar cortex layers are not clearly delineated here.

Quantitative RT-PCR of dissected PC and IHC studies prove, that VEGFR-2—as being expressed in all development stages, but significantly decreased in mature PC—is most likely to be the decisive receptor for VEGF, most likely controlling the growth of these neurons. A naturally similar expression of VEGFR-2 in the cerebellar roller tube slice cultures showed optimal conditions for the investigation of miR204-5p effects regarding VEGFR-2 expression.

### 2.3. sh-miR204-5p Construct Causes miRNA Overexpression and KDR Suppression in N2A Cells

Before using the vector containing miR204-5p for microinjection (Figure 4a), we tested its functionality in regards to *KDR* expression in neuronal tissue to ensure a high validity of our results. Therefore, Neuro 2A (N2A) cell lines were transfected with pLKO-miR204-DsRed and pLifeAct Tag red fluorescent protein (RFP) in comparison to mock transfection (MOC) and the expression values for miR204-5p and *KDR* were measured via qRT-PCR (Figure 4b,c). This experiment allowed us to gain important basic insights.

pLKO-miR204-DsRed produced a very clear and significant overexpression of miR204-5p copies (Figure 4b), while in the control groups (MOC and pLifeAct Tag RFP) the amount of miR204-5p copies remained low (Figure 4b). In turn, miRNA overexpression provoked a significant decrease of *KDR* in N2A cells transfected with the pLKO-miR204-DsRed vector. The expression of *KDR* reached approximately 19% compared to the controls (Figure 4c). This experiment first demonstrated a possible relationship between miR204-5p and VEGFR-2 density on the protein level in a neuronal cell line.

### 2.4. Impact of miR204-5p Overexpression on PC Dendritic Development

#### 2.4.1. Dendritic Growth of Juvenile and Mature PC in Response to VEGF (Control Groups)

To investigate the natural effects of VEGF on juvenile (group 2) and mature (group 3) PC, cells were injected simultaneously with pLifeAct Tag RFP and an enhanced green fluorescent protein (pEGFP-C1) and incubated for 24 h (Figure 5 a1,a2). The cerebellar slice cultures were then cultivated for a further 48 h with or without VEGF addition (Figure 5b1,b2). Then the total length of the PC dendrites was measured (*n* ≥ 18 per condition), and the relative dendritic growth (RDG) was calculated (Figure 5a3,b3). For PC of group 2, a significant increase of dendritic growth was measured under VEGF treatment conditions (RDG = 1.160 ± 0.039) compared to untreated neurons (RDG = 0.960 ± 0.019; Figure 5c, A + B). In contrast, incubation with (RDG = 1.060 ± 0.021) and without VEGF (RDG = 1.050 ± 0.021) does not lead to significant changes in the dendritic growth of PC in group 3 (Figure 5f, A + B). It is clear that VEGF has an influence on the development of juvenile but not mature PC dendrites, while the VEGF sensitivity of the PC decreases with the maturation of the cerebellum.

#### 2.4.2. VEGF Response of Juvenile PC Is Blocked by miR-204-5p Overexpression

To visualize the effect of miR204-5p in juvenile PC (group 2), cells were incubated at p1 + 14 div after simultaneous microinjection of pLKO-miR204-DsRed and pEGFP-C1 with or without VEGF addition for 48 h. The total dendritic length was measured before and after treatment (*n* ≥ 9 per condition) and the RDG was calculated. Under the influence of miR204-5p overexpression, juvenile PC did not show a significant positive effect of VEGF on dendritic growth within 48 h (RDG = 0.993 ± 0.033), nor did untreated PC (RDG = 1.019 ± 0.027; Figure 5c, C + D). Pre-treatment with miR204-5p thus led to an abolition of the growth stimulus of VEGF on juvenile PC, most likely due to a downregulation of *KDR* by the miRNA.

#### 2.4.3. miR-204-5p Overexpression Causes Dendritic Length Reduction in Mature PC

As in juvenile PC, we performed microinjection with pLKO-miR204-DsRed and pEGFP-C1 also in group 3 PC (p9 + 21div) to study the effects of miR204-5p overexpression in mature PC (Figure 5d–f). Before and after 48 h cultivation with or without VEGF the total dendritic length was measured and the RDG was calculated. While the dendrites of untreated PC remained unchanged in length (RDG = 1.040 ± 0.038), a significant decrease in dendritic length was observed in VEGF treated PC (RDG = 0.901 ± 0.023) (Figure 5f, C + D). Surprisingly, pre-treatment with miR204-5p in adult cells led to VEGF reducing the dendrite length of the PC. This effect can be explained by a loss of neuroprotective effects due to the interaction of miR204-5p with *KDR* and the blocking of VEGFR-1 via VEGF and is discussed in more detail in Section 3.2.

## 3. Discussion

Despite increasing therapeutic and scientific interest in VEGF and its steadily growing clinical relevance since its discovery, information on its effects in the central nervous system (CNS) and its role as a neurotrophic factor is still limited. Therapeutic interventions are currently focusing on VEGF’s vasculo–angiogenetic properties, for example in the treatment of various cancers [9,14], diabetic retinopathy [50], and age-dependent macular degeneration [10]. However, looking at the potential for clinical use of VEGF in brain injuries [29,51], shows the importance of gaining a deeper understanding of the regulatory mechanisms of VEGF in the CNS. Although some effects of VEGF on cerebellar cells have been studied [52,53], there is still little knowledge about its effects on the development and maturation of PC and the mechanisms of VEGF receptor regulation in this context. Since an age-dependent expression of VEGFR-2 in PC has already been shown [16,54], which also implies an age-dependent positive VEGF effect on PC growth, the current study clarifies to what extent the miRNA miR204-5p is involved in this process.

For this purpose, the effects of VEGF on juvenile and mature PC pre-conditioned by microinjection with miR204-5p imitates were analysed in the organotypic cerebellar roller tube section cultures. It was shown that miR204-5p exerts an inhibitory effect on the expression of VEGF pathways. The natural increase in miR204-5p during PC development [41] is most likely the cause of a reduction in VEGFR-2 expression, whereby adult PC do not react further with an increase in dendrite growth after VEGF application.

### 3.1. VEGFR-2 Expression Decreases during Maturation in Neonatal, Juvenile, and Mature PC

As the influence of VEGF on the nervous system has recently become increasingly the focus of research, the VEGF receptor concentration, especially of VEGFR-2, the most important VEGF acceptor in the CNS [17,19], has now been studied in many neuronal tissues. In addition to neuronal tissue of the hippocampus [55], neural stem cells [56] or frontal and temporal cortex [57,58], VEGFR-2 could also be detected in other non-endothelial tissue, such as astrocytes [59,60]. The influence of VEGF on cerebellar development was more often attributed to the expression of VEGF receptors in cerebellar cells [23,53]. In addition, lower concentrations of VEGFR-2 have already been detected in adult samples [61]. Consistent with this, several previous studies underlined a decrease of VEGF and VEGFR-2 levels during neuronal maturation in the human forebrain [54] and primary cerebellar slice cultures of mice [32]. Recently Herrfurth et al. [16] were able to demonstrate the age-dependent downregulation of VEGFR-2 by means of immunohistochemistry, in situ hybridisation, and qRT-PCR of laser-dissociated PC. The authors also show that this regulation is functionally correlated with an age-related neurotrophic effect of VEGF on PC dendrito–somitogenesis.

In the present study, these results were confirmed by qRT-PCR from individually dissected PC at three different stages of development. Here, we could demonstrate a significantly reduced expression of *KDR* in mature compared to juvenile development stages, while it was most frequently measured in neonatal PC. The barely detectable expression of the other VEGF (co-)receptors—especially *FLT4*, although it is generally assumed to play a role in cerebellar neuronal function [53]—suggests that these play only a minor role in promoting VEGF-induced processes inside PC. The expression levels of *FLT1* and *NRP2* remained invariably low throughout PC development, while the mRNA of *NRP1* fell to a much lower level, similar to the VEGFR-2 expression in growing PC. On the one hand, this supports the general classification of NRP-1 as a common and necessary co-receptor of VEGFR-2, which, therefore, can be suggested to be expressed on the cells in a manner similar to that of VEGFR-2 [62,63,64]. On the other hand, NRP-1—in contrast to the other low-expressed VEGF receptors, which remained unchanged—also influences PC development dependent on (and in symbiosis with) the main receptor VEGFR-2. 

The positive VEGFR-2 signals in our studies, both in cerebellar cryosections and in organotypic slice cultures, support our results concerning the age-dependent decrease of VEGFR-2 mRNA. While a strong positive receptor signal was detected in young and still in juvenile PC, the VEGFR-2 protein visibly decreased in mature slices and cultures. Even though the assessment of the dynamics of protein expression on the basis of signal intensities of an IHC staining can be prone to error, the accuracy of the results obtained here can be assumed with the help of other studies with comparable results [16,32,54]. As higher levels of VEGFR-2 in p9 and p15 compared to p30 were also confirmed at mRNA level, this allows us to discuss VEGFR-2 expression in a developmental context.

### 3.2. VEGF Effects on Dendritogenesis Depend on miR204-5p in Juvenile and Mature PC

Approximately 30 days after birth, the PC evolve from a star shape with perisomatic pre-dendrites to their final polarity by first forming main and later secondary branches to finally form their mature dendritic form with tertiary branches and numerous dendritic spines [47]. Organotypic slice cultures at the age of p1 + 14 div (group 2) were studied as representatives of an early developmental stage in which PC are still sensitive to VEGF. In contrast, older cultures (group 3, p9 + 21 div) represent the processes in adult PC, where no positive VEGF effects, e.g., on dendrite growth, are detectable. Using the method of microinjection, we were able to label individual PC in an organotypic environment and measure the total dendritic length to calculate the RDG after 48 h in two different maturation stages. The use of our plasmid pLKO-miR204-DsRed for microinjection allowed us to additionally induce an overexpression of miR204-5p in injected cells. The addition of VEGF to cells with or without overexpression of miR204-5p allowed us to reveal and visualize the inhibitory effect of miR204-5p on the VEGF signalling pathway compared to the control group.

These experiments confirm and extend the results of Herrfurth et al. [16] regarding a positive VEGF effect on dendritogenesis in juvenile PC, but not in adult PC. Individual labelling of living PC by microinjection allowed direct quantification of VEGF-induced changes in PC morphology and showed additive effects of dendritogenesis for juvenile PC after VEGF exposure. 

However, our data on mature PC, where no significant increase in PC dendritogenesis was found, suggest that the positive VEGF effects on PC are age-related. Mature PC dendrites are apparently no longer sensitive to VEGF, while juvenile, not yet fully developed PC respond to the application of VEGF with positive dendritogenesis. The decrease in VEGF sensitivity in adult neurons has been similarly demonstrated in the growth cone of dorsal root ganglia in PNS [25]. From the sum of our previous data, two interim results can be derived. First, VEGF is a very potential stimulating factor in PC development, promoting dendritogenesis, especially in the early stages of development. Secondly, our results of the immunohistochemistry and qRT-PCR suggest that the key to explaining age-related VEGF effects on PC lies in how the expression of the corresponding receptor VEGFR-2 is modulated. 

The answer to this was provided by our results of PC measurements with miR204-5p overexpression before and after VEGF addition (Setup D, see Section 4.7.2). The previously observed growth-promoting effect of VEGF on dendritic length in usually sensitive juvenile PC was completely suppressed by miRNA overexpression in cells preconditioned with miR204-5p. Therefore, the present results demonstrate an inhibitory effect of miR204-5p on the VEGF pathway in young developing PC for the first time. This, in turn, leads to the conclusion that the upregulation of miR204-5p, previously identified by Pieczora et al. [41], in the context of PC development must indeed be considered as the cause of VEGFR-2 downregulation—and, thus, as a critical factor of naturally declining VEGF sensitivity in adult PC. Overexpression of a pLifeAct-RFP vector showed no effect on dendrite length with or without VEGF. A current limitation of the study is that no vectors with a shuffled sequence were used as controls. It remains to be verified whether such a vector has an influence on the VEGF signalling pathways. However, our experiments in N2A cells also show a clear link between miR204-5p overexpression and the suppression of *KDR.* Transfection of pLKO-miR204-DsRed induced a significant decrease of *KDR* expression compared to un-transfected N2A cells. Since the signalling pathways and molecular interactions observed in N2A cells at the mRNA level in the current work are transferable to other neuronal cells [65,66], miR204-5p can thus also be assumed to be a crucial VEGFR-2 suppressor in PC. An inhibitory effect of miR204-5p on VEGF pathways has also been demonstrated in other tissues, e.g., in corneal neovascularization or in blood cells and HEK (human embryonic kidney) cells in response to hypoxia [36,43]. However, a possible interaction between miR204-5p and VEGF itself, which is possible according to target prediction and hypoxia trials in tilapia [36,42], can be excluded as the cause of the effect found, since organotypic slice cultures were treated with exogenous VEGF-A165 protein and miR204-5p could therefore not interact with VEGF gene products [35,38]. 

For the mature phase, in which the growth of the PC usually completed, the VEGFR-2 density on the cells has already decreased, and the PC are no longer sensitive to VEGF, we did not actually expect any effect of miR204-5p. However, in fact we also saw in adults that miR204-5p overexpression with simultaneous application of VEGF for 48 h led to a reduction in dendritic length. The importance of at least a basal level of VEGFR-2 expression also in adults to stabilise the morphology and extension of the dendrites can be speculated based on these observations. It is also possible that the balance of interactions with VEGFR-1 plays a role, which is unlikely to be influenced by miR204-5p in adult PC [42]. In neuronal tissue, VEGFR-1 is thought to have neuroprotective effects mediated by PIGF. In vivo, this effect is also mediated in parallel by the VEGF/VEGFR-2 interaction, whereas VEGF that bind to VEGFR-1 does not cause any change in the gene expression patterns of the corresponding cell, even though it has a higher affinity to VEGFR-1 than for VEGFR-2 [19,20]. With regard to our slice cultures pre-conditioned with miR204-5p, however, we can assume a loss of VEGFR-2 on the PC surface, so that neuronal survival is only guaranteed via PIGF and VEGFR-1, yet not leading to any noticeable changes in the dendritic length of the investigated PC within 48 h. In cells treated with VEGF this situation obviously changed remarkably. While PIGF and VEGF did not previously compete for binding to VEGFR-1 [67], the balance changed significantly with the addition of VEGF, which now occupied the VEGFR-1 binding sites and consequently no longer allowed PIGF to have a neuroprotective effect via VEGFR-1 anymore [68,69]. Since cell shrinkage is usually one of the first signs that PC are beginning to age [70], we put the hypothesis that the overexpression of both miR204-5p and VEGF eliminated the neuroprotective effects of VEGFR-1 and -2 in mature PC, which inevitably initiated the ageing process and, therefore, a reduction of total dendritic length in the affected cells. Similar results in response to miR-204-5p overexpression have already been found in dopaminergic neurons [45]. In the present study, however, ageing effects were only visible as a result of additional VEGF treatment. To clarify these findings, further investigations will be necessary.

These data finally allow us to design a model of VEGF-driven development of maturing PC under control of miR204-5p. Within the first days of development, the neurotrophic and dendritogenetic effects of VEGF are necessary for the maturation of PC, which therefore express VEFGR-2 in a high amount. This is made possible at the juvenile stage by a low miR204-5p concentration [41]. In mature PC, the expression of VEGFR-2 is significantly reduced by the upregulation of miR204-5p [41], and the PC are therefore insensitive to VEGF growth signals. A small amount of remaining VEGFR-1 and 2 nevertheless still ensures neuroprotection of the PC. Thus, since the importance of VEGF as a neuronal growth factor constantly decreases during PC development, i.e., while the cell has already formed dendrites, the expression of VEGFR-2 also decreases during maturation. This process is controlled by a gradual increase in miR204-5p [41], which post-transcriptionally inhibits the *KDR* gene, resulting in reduced production of the VEGFR-2 protein. In conclusion, our present study shows that miR204-5p is at least a key modulator of VEGF sensitivity of PC, which most likely regulates VEGFR-2 expression. Further experiments, in which interception of miR204 is expected to re-enable a growth stimulus in PC by VEGF, may provide definitive evidence that miR204-5p plays a key role in PC development.

### 3.3. New Possibilities Regarding the Therapy of Neuronal Pathologies?

The therapeutic potential of VEGF—its antibodies, and inhibitory drugs—was recognised, and is already widely used in clinical practice. The areas of application are manifold and range from ophthalmological interventions in age-related macular degeneration or diabetic retinopathy [10,50] to tumour treatment against, e.g., breast cancer [9,14]. Recently, there have been major efforts to extend the field of application to neurological diseases in which VEGF has been shown to be involved in the development of the disease. For example, in diseases such as Parkinson’s disease [71], Alzheimer’s disease [72], or Amyotrophic lateral sclerosis (ALS) [73], there is hope that VEGF-based therapy can help to improve treatment. Great hopes are also placed in the therapeutic application of VEGF–VEGFR-modulations for spinocerebellar ataxia 1 (SCA1) [32], as well as in traumatic brain injuries such as stroke [31,51] or epilepsy [30]. Many of these neurodegenerative diseases are associated with a disturbed genesis of dendrites [74]. The results of our study could show that special attention must be paid to the regulation of VEGFR expression during development. In mature PC, miR-204-5p is responsible for the decrease of VEGFR-2 expression and thus for VEGF insensitivity in these neurons. Therefore, inhibition experiments of miR204-5p might be of interest in other neurons to counteract the lost sensitivity of adult neurons to VEGF. If inhibition of miR204-5p, e.g., by antagomiRs [75,76], were therapeutically possible and morphological plasticity in adult PC could be positively triggered by VEGF, this would open many new possibilities in the treatment of neurological pathologies.

## 4. Materials and Methods 

### 4.1. Tissue Cultures

All procedures were conducted under established standards of the German federal state of North Rhine Westphalia, in accordance with the European Communities Council Directive 2010/63/EU on the protection of animals used for scientific purposes. 

Primary cerebellar slice cultures (stages p15 and p30) were prepared as described in previous protocols [16,77,78]. In brief, Wistar rats of male and female pups at postnatal day 1 and 9 (p1, p9) were decapitated. After dissection of the brain, cerebella were separated from the forebrain and brainstem with a razor blade and stored in cooled Hanks-medium (H1641; Sigma-Aldrich, St.Louis, MO, USA) containing 2 mg/mL penicillin (P3032; Sigma-Aldrich) and 0.5 mg/mL glucose (A1349, 1000; Biochemica, Billingham, UK). With help of a binocular microscope, the pia mater and small superficial blood vessels were detached carefully from the isolated cerebella. Cerebella were cut into parasagittal slices with a thickness of 275 µm on the McIlwain tissue chopper and brought into cooled Gey’s balanced salt solution (G9779; Sigma-Aldrich). After careful separation, single slices were applied to collagen-coated (C7661; Sigma-Aldrich) glass coverslips (32 mm; Kindler, Bobingen, Germany), fixed with 10 µL chicken plasma (P3266; Sigma-Aldrich) and 10 µL thrombin (605157; Calbiochem, Sigma-Aldrich) and left at room temperature (RT) until coagulation. Each cover slip was then placed into a roller tube filled with 1.3 mL warm nutrient medium (basal medium eagle (BME, B1522; Sigma-Aldrich), 25% heat-inactivated foetal horse serum (S9135; Biochrom, Stockelsdorf, Germany), 25% Hanks-medium (H1641; Sigma-Aldrich), 10% 6.5 mg/mL glucose, 0.01 g/mL penicillin (P3032; Sigma-Aldrich), 0.01 g/mL L-glutamine (G7513; Sigma-Aldrich), and 25 ng/mL nerve growth factor (NGF-7S, N0513; Sigma-Aldrich)). Following that, the roller tube slice cultures were maintained slowly rotating in a roller-drum incubator at 37 °C.

The roller-tube nutrient medium was replaced twice a week for a period of up to 14 div (p1 cerebella) and 21 div (p9 cerebella). After, 10 div slices were maintained with nutrient medium possessing a reduced concentration of fetal horse serum (15%). To reduce the number of fibroblasts on the slices, cultures from older rats (p9) were treated with a mitosis inhibitor mixture (0.33 mM Uridine, 0.33 mM Cytosine-β-D-arabinofuranoside hydrochloride, 0.33 mM 1-(2-Desoxy-β-D-ribofuranosyl)-5-fluorouracil; U3003, C6645; Sigma-Aldrich; 21555, Serva, Heidelberg, Germany) after 4 div for 24 h, and afterwards cultivated with nutrient medium again as described before.

### 4.2. Experimental Groups

All experiments were performed at three different stages of cerebellar development: neonatal (group 1: p9) as well as juvenile (group 2: p15) to investigate developing PC and mature (group 3: p30) to investigate PC in fully outgrown dendritic shape (Table 1).

Since slice cultures need to flatten at least two weeks in vitro before being able to be used for the microinjection technique, we chose stage p1 plus 14 div PC as the juvenile experimental group (group 2) and stage p9 plus 21 div PC as the mature experimental group (group 3). Therefore, a microinjection study of group 1 was not feasible due to poor injection conditions in cultures below 14 div (Table 1). 

### 4.3. Cryosections of Rat Cerebella

To obtain cerebellar cryosections, p9, p15, and p30 Wistar rats have been decapitated and cerebella were collected in 4% paraformaldehyde (PFA) in phosphate buffered saline (PBS; 18912-014; Gibco). Tissue was post fixated for three days. Following that, cerebella were isolated and fixated with 4% PFA in PBS at 4 °C overnight. Cerebella for immunohistochemistry were washed with PBS the next day, transferred into 30% sucrose (A22115000; AppliChem, Darmstadt, Germany) in PBS for two days at 4 °C and finally deep-frozen in isopentane at −50 °C. Cerebella for laser microdissection were directly deep-frozen in isopentane at −50 °C after PBS washing. In both cases, samples were stored at −80 °C until use. For the cryosection process, deep-frozen cerebella were cut using a cryostat (Leica CM 3050 S; chamber temperature −18 °C, stage temperature −20 °C) after fixation on a stage using tissue freezing medium (no. 14020108926; Leica, Wetzlar, Germany). Cryosections for immunohistochemistry (40 µm thick) were applied on glass microscope slides (J1800AMNZ; Menzel-Gläser, Braunschweig, Germany) while cryosections for laser microdissection (10 µm thick) were placed on polyethylene naphthalene (PEN) membrane steel frame slides (No. 11505151; Leica). For laser microdissection, all operational steps were performed under sterile RNase free conditions [41].

### 4.4. Immunohistochemistry (IHC)

After cultivation of rat cerebellar slice cultures for the appropriate time, the cerebellar slices were fixated with 4% PFA in PBS for 1 h and bathed in PBS three times for 10 min. The slices were permeabilised with 1% Triton-X-100 (T8532; Sigma-Aldrich) in PBS for 15 min and washed with PBS again three times for 2 min. To avoid non-specific binding, goat serum (1:50 in PBS, G9023; Sigma-Aldrich) was added for 30 min. After another short PBS washing step, the slices were incubated with primary anti-calbindin-D28K antibodies (mouse, 1:500, C9848; Sigma-Aldrich) diluted in PBS at 4 °C overnight to specifically label PC. On the next day, the slices were rinsed with PBS three times for 2 min and two times for 10 min before secondary anti-mouse immunoglobulin G IgG TRITC-coupled antibodies (goat, 1:500, T5393; Sigma-Aldrich) diluted in PBS were applied for 2–3 h at RT. After washing with PBS three times for 2 min and two times for 10 min, primary antibodies against VEGFR-2 (rabbit, 1:300, 39638; Abcam, Cambridge, UK) diluted in PBS were added and the slices were incubated at 4 °C overnight. After washing with PBS three times for 2 min and two times for 10 min, secondary Alexa Fluor 488-coupled IgG antibodies against rabbit (donkey, 1:500, A21206; Life Technologies, Carlsbad, CA, USA) diluted in PBS were applied on the slice cultures for 2–3 h at RT. Subsequently washing with PBS three times for 2 min and two times for 10 min was performed and nuclear counterstaining was done with bisBenzimide Hoechst 33,342 trihydrochloride (1:1000, B2261; Sigma-Aldrich) diluted in PBS for 20 min. Finally, the slice cultures were rinsed with PBS three times for 10 min and embedded in fluorescence mounting medium (FluoroShield, F6937-20ML; Sigma-Aldrich) on microscope slides (J1800AMNZ; Menzel-Gläser, Braunschweig, Germany) and stored at 4 °C until use. 

For p9, p15, and p30 rat cerebellar cryosections the same immunostaining protocol was used. The only difference, in this case, was that the embedding was done with fluorescence mounting medium under a coverslip (S3023; Dako, Jena, Germany) covering the stained cryosection applied to the microscope slide.

### 4.5. RNA Expression Level Analysis (Groups 1 + 3)

#### 4.5.1. Methylene Blue Staining

The cooled probes were stained with a few drops of dye containing 1% methylene blue, 1% azure II, and 1% Borax in DEPC-treated water. After application, the blue dye was removed immediately using DEPC-treated water. The sections were air-dried and stored at RT before laser microdissection.

#### 4.5.2. Laser Microdissection (LMD)

The slides were microdissected with the LMD6500 system (Leica) as already described by Pieczora et al. [41] using the following settings: 40× objective, power 23, aperture 21, speed 23 and specimen balance 18. With each session, about 1,000,000 µm^2^ of PCs was lasered and collected in non-adhesive 0.5 mL tubes by marking PC separately for the laser beam. After microdissection, 40 µL of lysis solution (AM1931; Thermo Fisher, Waltham, MA, USA) was added to each sample. All samples were stored at −80 °C until further usage. In total, about 4,000,000 µm^2^ of PC from four different rat cerebella were collected for each experimental group (1–3).

#### 4.5.3. Total RNA-Isolation 

Total RNA-extraction from the collected PC-enriched tissue was performed with the RNA-aqueous-micro total RNA isolation kit (AM1931; Invitrogen, Carlsbad, CA, USA) according to the manufacturer’s protocol. Briefly, the samples containing about 2,000,000 µm^2^ of PC in 100 µL lysis solution were incubated at 42 °C for 30 min and prepared with 3 µL laser capture microdissection (LCM) Additive and 100% ethanol according to the protocol. After filtration and several washing steps using the filter cartridge and wash solutions, the small and large RNA was eluted two times with preheated Elution solution from the filter. For qRT-PCR preparation, the samples were finally treated with DNase I and DNase inactivation reagent, according to the protocol to remove contaminating genomic DNA. For the last step, the eluted total RNA (about 25 µL) was transferred into a fresh tube and stored at −80 °C until further usage.

#### 4.5.4. cDNA Synthesis + qRT-PCR

cDNA synthesis of the mRNA was performed using GoScript Reverse Transcription Mix Oligo(dT) (A2790, Promega, Fitchburg, WI, USA), following the manufacturer’s protocol. cDNA concentration was measured photometrically (Genova Nano; Jenway, Stone, UK) and diluted to 200 ng/µL cDNA with nuclease free water (P1193, Promega). All samples were stored at −20 °C until use.

The qRT-PCR was performed with GoTaq qPCR Master Mix (A6001, Promega) according to the manufacturer’s protocol using 800 ng of cDNA per reaction and 0.7 µM of each Primer (forward + reverse) for each well. Primer sequences used are: *FLT1* (forward GTG AAG AGT GGG TCG TCA TTC, reverse CTA TGG TTT CCT GCA CCT GTT); *KDR* (forward TCC CAG AGT GGT TGG AAA TG, reverse ACT GAC AGA GGC GAT GAA TG); *FLT4* (forward CTG GAC ACC CTG TAA GAC ATT T, reverse AGT GGT CAC CTC CTT CCA); *NRP1* (forward AGA TCG CCT ACA GTA ACA ATG G, reverse CTT GTG GAG AGA GGT GTA AAG G); *NRP2* (forward GGC TTC TCA GCA CGT TAC TAT T, reverse TGA GGC ACT GAT CTG TTC ATT AG) and the housekeeping gene *GAPDH* (forward ACT CCC ATT CTT CCA CCT TTG, reverse CCC TGT TGC TGT AGC CAT ATT). A quantitative PCR was performed on a CFX Connect Real Time PCR Detection System (Bio-Rad, Hercules, CA, USA) and analysed after the PCR run using the CFX Manager Software to obtain melting curves showing single PCR products.

#### 4.5.5. Statistical Analysis

Expression levels for the genes of interest and for housekeeping gene *GAPDH* were measured in three independent PCR runs. Fold change of expression was calculated using the 2^(–ΔΔCt)^ method. The relative expression levels of validated mRNAs were all calculated in relation to *KDR* expression in p9 PC and compared using an unpaired two-tailed students *t-test*. For statistical analysis, Microsoft Excel (Microsoft 365; Microsoft, Redmond, WA, USA) and Graph Pad Prism (version 6.01; Graph Pad, San Diego, CA, USA) were used. 

### 4.6. Vector Preparation

#### 4.6.1. Creating the sh-miR204-5p Cassette

To create the sh-miR-204-5p cassette, oligonucleotides (5′-CCG GAG GCA TAG GAT GAC AAA GGG AAC TCG AGT TCC CTT TGT CAT CCT ATG CCT TTT TTG -3′, 3′- AAT TCA AAA AAG GCA TAG GAT GAC AAA GGG AAC TCG AGT TCC CTT TGT CAT CCT ATG CCT-5′) were annealed with 5× T4-DNA Ligase Buffer (15224041; Invitrogen) in a thermal cycler (PTC-200; MJ Research, St.Bruno, QC, Canada) gradually cooling from 99 °C to 6 °C within 70 min to avoid a short hairpin configuration of the hybrids. The sample was stored at −20 °C until use.

#### 4.6.2. Cloning the sh-miR204-5p into the Backbone

The backbone vector (Tet-pLKO-puro, Plasmid #21915; Addgene, Watertown, MA, USA) was digested doubly with endonucleases EcoRI-HF (R3101S; New England BioLabs, Ipswich, MA, USA) and AgeI-HF (R3552S; New England BioLabs) in CutSmart Buffer (B2704S; New England BioLabs), according to the manufacturer’s recommendations for 1 h at 37 °C to remove the 1.8 kb stuffer. Following that, the fragments were separated in Tris-acetate-EDTA (TAE) gel containing 1.5% agarose (9012-36-6; Carl Roth, Karlsruhe, Germany) at 140 V for 30 min. For further usage, the 8.9 kb AgeI/EcoRI band (marker: 1 kb plus Ladder (10488085; Invitrogen)) was excised and DNA was extracted with the PeqGOLD Gel Extractions Kit (732-2777; PeqLab, Erlangen, Germany) following the manufacturer’s protocol and DNA concentration was measured photometrically via NanoDrop (ND 1000; PeqLab).

To insert the double stranded sh-miR204-5p sequence into the open backbone, ligation was performed overnight at 14 °C with T4-DNA Ligase and 5× Ligation Buffer. After purification of nucleic acids via phenol-chloroform (PC8) extraction and ethanol (ETOH) precipitation of the ligation product, the DNA concentration was measured photometrically (ND 1000, PeqLab). For targeted sequencing, the plasmid has been prepared using the GenomeLab DTCS Quick Start Kit (PN 608120; Beckman Coulter, Krefeld, Germany), according to the manufacturer’s protocol using a modified thermal cycling program: 20 s at 96 °C, 20 s at 50 °C, and 4 min at 60 °C (40 cycles) with 110 °C lid temperature (Mastercycler gradient; Eppendorf). The primers used were pLKO-MB primers (sense GAC TGT AAA CAC AAA GAT ATT AG and antisense TAA TTC TTT AGT TTG TAT GTC TG). A purification of the cycling product was performed then using SPRI CleanSEQ Magnetic Beads (Agencourt CleanSEQ Kit, A29154; Beckman Coulter) following the manufacturer’s protocol. The sequencing process was performed with a CEQ-8000 Analyser (BE-CEQ; Beckman Coulter) and analysed using the GeneScreen software (Insilicase; insilicase.com) showing that the correct sequence of sh-miR-204-5p could be found within the plasmid.

#### 4.6.3. Replacing the Puro Cassette by DsRed

For the second cloning step, both the new plasmid containing the sh-miR204-5p cassette and puromycin (pLKO-sh-miR204-5p-puro), and a donator plasmid (pLKO-U6-DsRed) were double-digested with BamHI-HF (R3136S; New England BioLabs) for 30 min at 37 °C and SfiI (R0123S; New England BioLabs) for 30 min at 50 °C with CutSmart Buffer according to the manufacturer’s recommendations. The fragments were separated within a TAE gel containing 1.5% agarose and the 1.3 kb DsRed from the donator pLKO-U6-DsRed well as the 7.6 kb BamHI/SfiI band (marker: 1kb plus Ladder) from the backbone were excised and DNA was extracted using the PeqGOLD Gel Extractions Kit (732-2777; PeqLab, VWR Chemicals) following the manufacturer’s protocol. DNA concentration was measured photometrically (ND 1000, PeqLab) and ligation was performed at RT for 1 h with T4-DNA Ligase and 5× Ligation Buffer. 

#### 4.6.4. Transformation, Selection and Verification

After PC8 extraction and EtOH precipitation, the ligation mixture was transformed into *Escherichia coli* (*E. coli* OneShot STLB-3 bacteria; Life Technologies) using standard electroporation procedure (Electroporator 2510; Eppendorf, Hamburg, Germany). Transformed cells were shaken for 1 h with 190 rpm at 37 °C in lysogeny broth (LB) medium (LB-broth, L2542; Sigma-Aldrich). The cells were plated on LB-agarose-agar (LB-broth; SELECT Agar, 30391049; Invitrogen) containing 100 µg/mL carbenicillin (C1389; Sigma) and incubated at 37 °C overnight. The next day, grown colonies were picked and incubated at 37 °C and 190 rpm for 4 h in 200 µL LB medium each in a 96-well plate and then centrifuged at 2500 rpm for 3 min (Multifuge 3 SR; Heraeus, Hanau, Germany). The pellet was resuspended in water and 1 µL of each bacteria suspension was added to a PCR plate well containing reaction mixture (1.5 µL 10× TrisA buffer, 0.6 µL MgCl₂ (50 mM), 0.3 µL dNTPs (10 mM), 0.5 µL primers (sense + antisense; 10 µM), 10.3 µL DEPC water and 0.3 µL Taq-polymerase (5000 units/µL per well)) for direct Bac PCR. The PCR was performed in a thermocycler first at 95 °C for 3 min following 35 cycles of 95 °C for 30 s, 55 °C for 30 s and 72 °C for 30 s. Primers used were DsRed-sense (AGC TGG ACA TCA CCT CCC ACA ACG) and pLJM-antisense (CCA ATG ACT TAC AAG GCA GC). 

Following that, 5 µL of each PCR product were tested for positive clones using a TAE gel containing 1.5% agarose with 1 kb plus Ladder as a marker. Plasmid DNA minipreparations were made from overnight cultures using PureYield Plasmid Miniprep System (A1223; Promega), following the manufacturer’s protocol. The DNA concentration of the sample was measured photometrically and a test restriction of the mini prep was performed showing the expected fragments. 

Finally, 10 µL of bacteria suspension were incubated in 150 mL LB medium and 150 µL carbenicillin overnight at 37 °C and 190 rpm and a midi-preparation was performed the subsequent day using PureYield Plasmid Midiprep System (A2492; Promega). The correct sequence was verified via standard sequencing and the resulting plasmid was stored at −20 °C until use. In the following, our product is called pLKO-miR204-DsRed.

#### 4.6.5. Plasmid Testing

Before use, the finished plasmid was tested in two independent test series. At first, it was investigated whether an overexpression of the desired sh-RNA miR-204-5p could be detected after its transfection into cell cultures. For this purpose, 100,000 neuroblastoma cells (Neuro-2A, N2A) cells per well were plated into a 6-well plate and incubated for 24 h. The plasmids were then transfected into the N2A cells using pLKO-miR204-DsRed, a red fluorescent protein plasmid (pLife-Act Tag RFP, 60102; Ibidi, Gräfelfing, Germany) to exclude effects of introducing a vector unrelated to VEGFR-2 and the pure transfection reagent (MOC) as a control for two wells each. 

For transfection, a 100 µL total volume of electrical conductivity EC buffer containing 1 µg of the respective DNA was prepared for each of the two DNA samples using the Effectene Transfection Reagent (301425, Qiagen, Hilden, Germany) according to the manufacturer’s protocol. The cells were then incubated for a total of 48 h at 37 °C, 95% humidity, and 5% CO_2_. In order to determine the concentration of miR-204-5p in the different approaches, the total RNA had to be isolated from the cells. After a short washing step with PBS, the cells were treated with trypsin/EDTA (59417, Sigma-Aldrich) for 3 min at 37 °C. After a maximum incubation time of 5 min, 2 mL nutrient medium was added to each well, the cells were centrifuged for two minutes at 800 rpm. The supernatant was discarded, and the cells were then re-suspended in 300 µL ML lysis buffer (NucleoSpin miRNA Isolation Kit, REF740971.50; Macherey-Nagel, Düren, Germany). After that, RNA isolation was performed following the instructions of the miRNA isolation kit according to the manufacturer’s recommendations.

qRT-PCR to quantify the miR-204-5p expression in the cells after transfection and statistical evaluation was finally performed according to the protocol in Section 4.5.4. The reverse transcription was performed with 500 ng tRNA according to the miRCURY LNA RT Kit (339340; Qiagen); the qPCR was performed using primers against miRNA-204-5p (YP00206072, Qiagen) and U6 (YP00203907, Qiagen) as a control gene. The expression level of miR-204-5p could be compared natively (MOC) under influence of an unrelated vector (RFP) and under addition of the plasmid (pLKO-miR204-DsRed). 

The second series of experiments were designed to decipher whether overexpression of miR-204-5p can influence *KDR* expression in neuronal tissue. N2A cells were plated analogous to the first series of experiments and transfected with the plasmid pLKO-miR204-DsRed and MOC. After successful RNA isolation, reverse transcription of 1 µg RNA was performed using the qScript cDNA SuperMix (95048-025; Quantabio, Beverly, MA, USA) and qPCR was performed with GoTaq qPCR Master Mix and primers against *KDR* and *GAPDH* as housekeeping gene as already described in Section 4.5.4. The expression of *KDR* in the neuronal cell line could then be compared with (pLKO-miR204-DsRed) and without (MOC) overexpression of miR-204-5p.

### 4.7. Microinjection and Morphometric Analysis (Groups 2 + 3)

#### 4.7.1. Microinjection Process + Setup

By applying the microinjection technique, the plasmid containing miR-204-5p was transferred into a single juvenile PC (group 2: p1, 14 div) and mature PC (group 3: p9, 21 div) of cerebellar slice cultures as described before [16,77]. For injection, 2 µL of the plasmid mixture (see below) were filled into self-pulled sterile capillaries (Micropipette Puller Model p-1000; Sutter Instrument, Novato, CA, USA) with filament (borosilicate glass capillaries: 1.5 mm/0.2 mm, Nr.1403512; Hilgenberg, Malsfeld, Germany). PC were imaged on an inverted microscope equipped with long-distance phase-contrast optics (Axiovert 35; Zeiss, Oberkochen, Germany) while kept at 37 °C in nutrient medium on a heating stage for the whole microinjection process. The pressure injection devices (InjectMan NI2 and FemtoJet; Eppendorf, Hamburg, Germany) were set to inject with 70–80 hPa for 0.6 s while constant pressure was defined as 50–60 hPa. Following the microinjection, the slice cultures were transferred into sterile TC dishes (83.3900; Sarstedt, Nümbrecht, Germany) with fresh nutrient medium and further incubated at 37 °C in an atmosphere of humidified 95% air and 5% CO_2_ for at least 24 h to receive a sufficient fluorescent signal.

In order to be able to display the PC with all their dendrites, pLKO-miR204-DsRed was injected simultaneously with a green fluorescent protein plasmid (pEGFP-C1). Thus, a mixture of pLKO-miR204-DsRed and pEGFP-C1 was prepared in the ratio 4:1 dissolved in sterile filtered water. The mixture contained 4.4 ng/µL of pLKO-miR204-DsRed and 1 ng/µL of pEGFP-C1. For the purpose of comparison at a later stage, controls were performed with pLifeAct Tag RFP and pEGFP-C1 in the same ratio.

#### 4.7.2. In Vitro Treatment and Morphometric Analysis of PC

After at least 24 h of incubation past microinjection, confocal laser scanning microscopy (CLSM) (LSM 800; Zeiss) was performed with microinjected cells. All PC giving a sufficient signal were imaged using the ZEN microscope software (version 2.3, blue edition, Zeiss). This point of time was set as t = 0 h. In order to investigate the effects of miR-204-5p on the VEGF-response in juvenile (group 2) and mature PCs (group 3), injected slice cultures were incubated for 48 h in nutrient medium either treated with 0.1 ng/µL VEGF (SRP4365; Sigma) or remained untreated for 48 h. After incubation (t = 48 h), CLSM was performed again with both treated and untreated slice cultures, PC imaged at t = 0 h were identified and photographed for a second time. For each group (2 + 3), four experimental setups were used (A–D, Table 2). Each dendrite of the PC investigated was measured and all distances were added to give the full dendritic length of each cell at t = 0 h and t = 48 h. To give the relative dendritic growth (RDG) of each cell within the 48 h of treated or untreated incubation, dendritic length at t = 0 h was set as 100% (RDG = (t = 48 h)/(t = 0 h)). In total, statistical analysis was performed with ≥36 juvenile (group 2) and ≥36 mature (group 3) PC (*n* ≥ 9 for each experimental setup) taken from at least 70 rat cerebella. To make sure both plasmids of the injected mixtures were sufficiently read off by the cell, only PC showing a clear red and a green signal (t = 0 h or t = 48 h) were included into the analysis. For statistical analysis, Microsoft Excel and Graph Pad Prism were used.

## Figures and Tables

**Figure 1 ijms-22-02173-f001:**
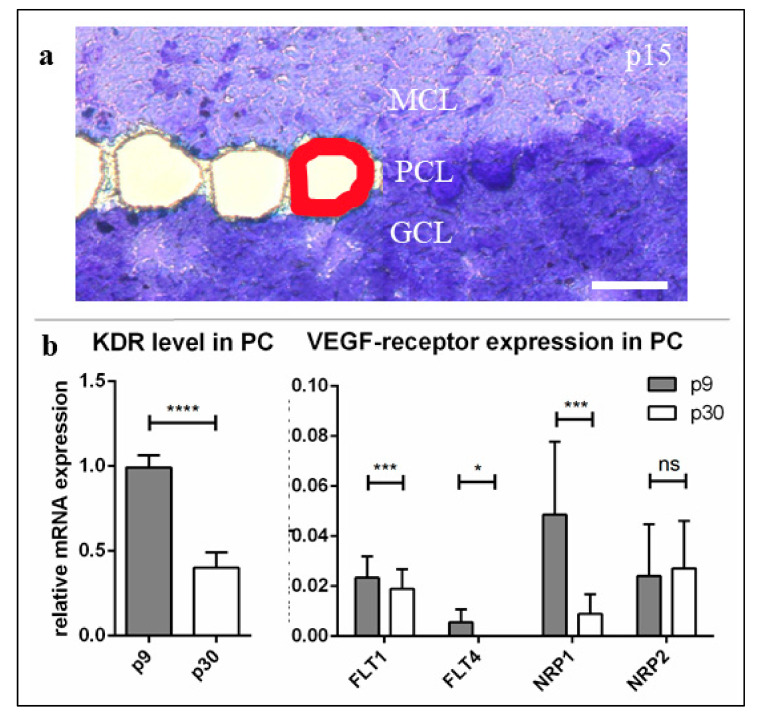
mRNA-expression levels of vascular endothelial growth factor (VEGF)-receptors in the course of Purkinje cells (PC) development. P9, (**a**) p9, p15, and p30 PCs were laser microdissected (red line) at 20× magnification and subjected to the method of qRT-PCR. (**b**) For relative quantification of *FLT1* (VEGFR-1), *KDR* (VEGFR-2), *FLT4* (VEGFR-3), *NRP1* and *NRP2*, the 2^(–ΔΔCt)^ method was conducted using *GAPDH* for normalisation. Relative mRNA expression is shown in relation to *KDR* at p9. Data are provided as means ± SEM and were tested for significance using Student’s t-test. Significant differences are indicated by ns (*p* > 0.05), * (*p* ≤ 0.05); *** (*p* ≤ 0.001), **** (*p* ≤ 0.0001), *n* = 3. Scale bar: (**a**) 50 µm.

**Figure 2 ijms-22-02173-f002:**
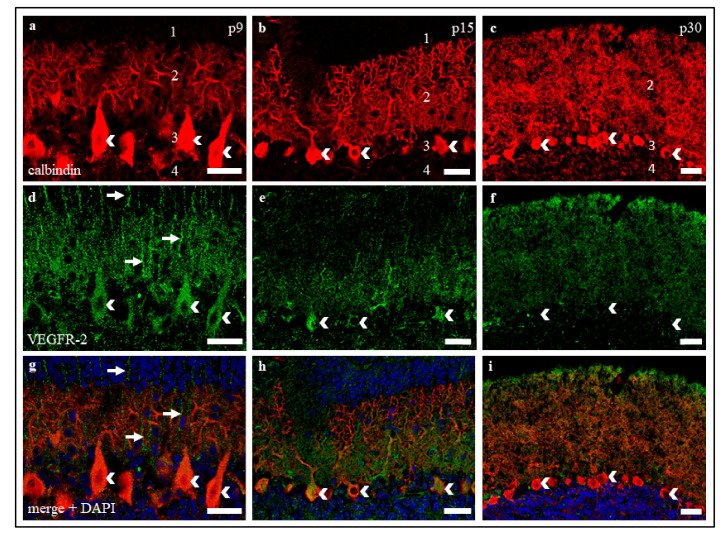
VEGFR-2 expression in cryosections of p9, p15, and p30 cerebella. (**a**–**i**) Immunostaining of calbindin positive PC (red), VEGFR-2 (green) and cell nuclei (blue) in cerebellar cryosections of rats at p9, p15, and p30. The images show that VEGFR-2 is localised in PC soma (white arrowheads) and dendrites. (**a**,**d**,**g**) At p9, VEGFR-2 could also be found in Bergmann glia fibres ascending to the external granular cell layer (eGCL) (white arrows). The same exposure settings were used for all exposures and no post-correction was applied. 1 = eGCL, 2 = molecular cell layer (MCL), 3 = Purkinje cell layer (PCL), 4 = GCL. Scale bars: (**a**,**d**,**g**) 50 µm; (**b**,**c**,**e**,**f**,**h**,**i**) 100 µm.

**Figure 3 ijms-22-02173-f003:**
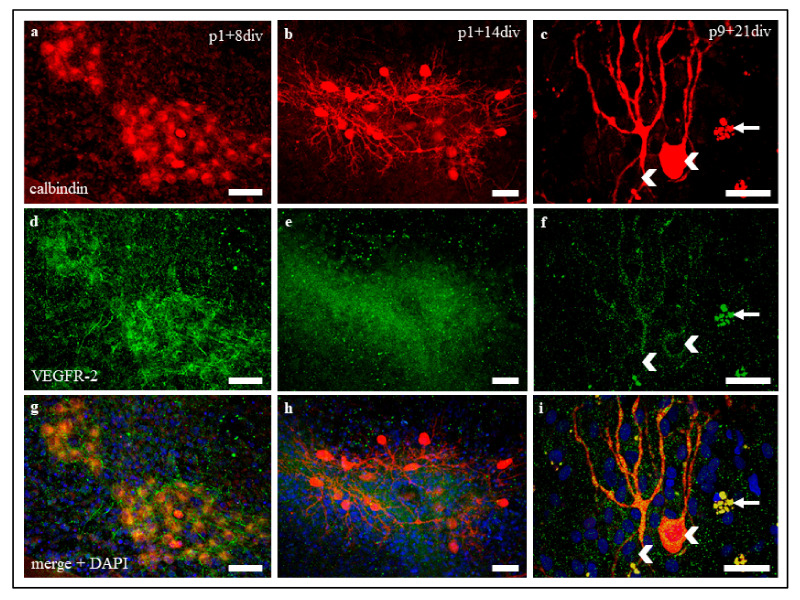
VEGFR-2 expression in organotypic slice cultures at p1 + 8 div, p1 + 14 div, and p9 + 21 div. (**a**–**i**) Immunostaining of calbindin positive PC (red), VEGFR-2 (green), and cell nuclei (blue) in organotypic cerebellar slice cultures at p1 + 8 div, p1 + 14 div, and p9 + 21 div. VEGFR-2 is detectable in PC soma and dendrites in all development stages. (**c**,**f**,**i**) Exemplary, two PC soma (white arrowheads) and some cell detritus (died PC, white arrows) are marked. The same exposure settings were used for all exposures and no post-correction was applied. Scale bars: 100 µm.

**Figure 4 ijms-22-02173-f004:**
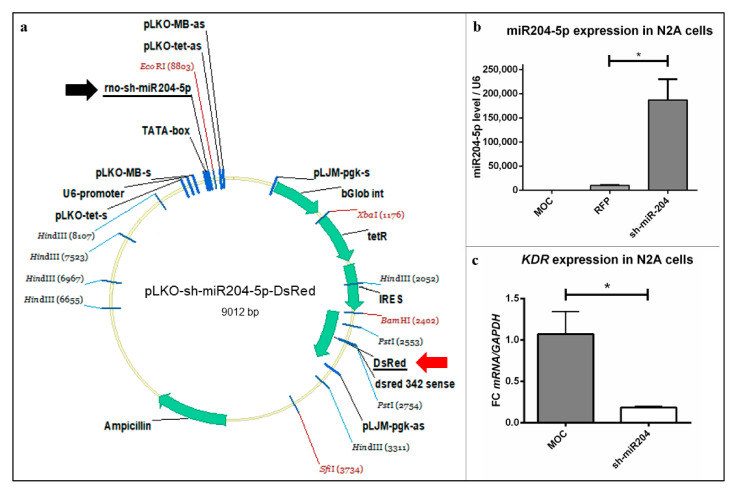
Vector map and testing of pLKO-miR204-5p-DsRed. (**a**) Vector used for microinjection contained the sh-miR204-5p sequence (black arrow), to investigate its effect on VEGF-triggered growth of PC, as well as DsRed (red arrow) to detect a successful injection of the plasmid into the PC. (**b**,**c**) N2A cells were transfected and processed for qRT-PCR. (**b**) For relative quantification of miR204-5p overexpression and (**c**) *KDR* expression level, the 2^(–ΔΔCt)^ method was accomplished using (**b**) U6 and (**c**) GAPDH as housekeeping genes. Controls: MOC—pure transfection reagent; red fluorescent protein (RFP)—LifeAct Tag RFP. Data are provided as means ± SEM. Significant differences are indicated by * *p* < 0.05; *n* = 3.

**Figure 5 ijms-22-02173-f005:**
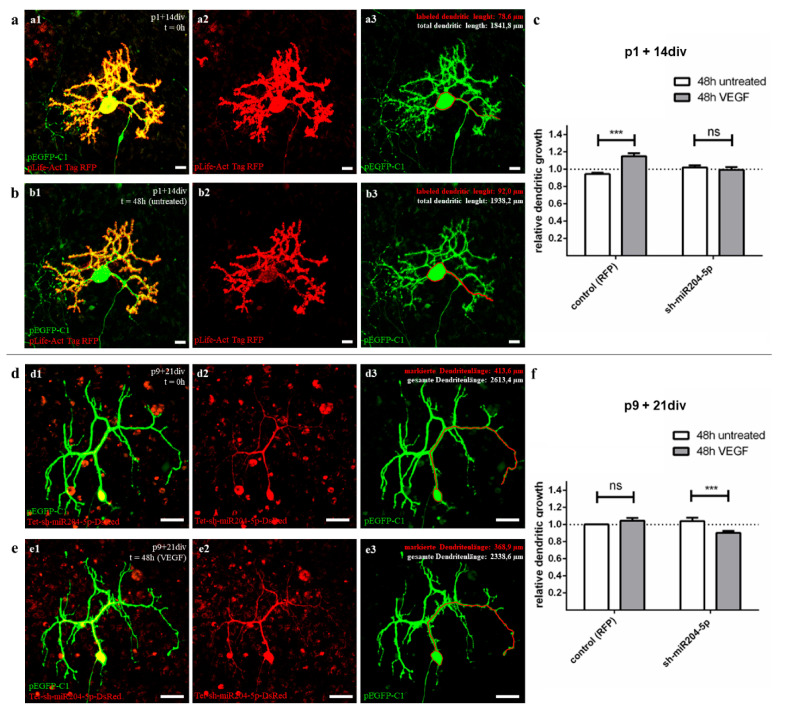
Morphometric analysis of miR204-5p effects on VEGF controlled dendritic growth in juvenile and mature PC. (**a**,**b**) Exemplary presentation of dendritic length analysis of PC at the age of p1 + 14div (group 2) injected with pEGFRP-C1 (green) and pLife-Act Tag RFP (red) before (**a**) and after 48 h untreated incubation (**b**). (**d**,**e**) Example of a PC at age p9 + 21div, injected with pEGFRP-C1 (green) and pLKO-miR204-DsRed (red) before (**d**) and after 48 h incubation with 0.1 ng/µL VEGF (**e**). The total dendritic length was measured for each cell that showed both sufficient red and green signal (one dendrite is shown in red for each cell before and after treatment as an example), and relative dendritic growth (RDG) was calculated. (c f) Effects of miR204-5p on VEGF-controlled dendritic growth in juvenile (**c**) and mature PC (**f**). Changes in dendritic growth are given in relative ratio (RDG = (dendritic length at t = 0 h)/(dendritic length at t = 48 h)); data are given as mean values ± SEM. Cells injected with pLife-Act Tag RFP only are defined as controls representing the natural VEGF reaction compared to the sh-miR204-5p injected PC. Significant differences are indicated by *** *p* < 0.0001, ns *p* > 0.05; *n* ≥ 9 (each bar). Scale bars: (**a**,**b**) 20 µm (**d**,**e**) 50 µm.

**Table 1 ijms-22-02173-t001:** Overview of age groups and experiments. Listed are the developmental stages of (PC used in this research work, including the corresponding slice culture incubation times and the applied experimental protocols. IHC = immunohistochemistry; ✓ = slice cultures and in vivo; SC = slice cultures only; X = not possible.

Group	In Vivo	Cultures	IHC	mRNA Quantification	Injection
(1) neonatal:	p9	p1 + 8div	✓	In vivo	X
(2) juvenile:	p15	p1 + 14div	✓	In vivo	SC
(3) adult:	p30	p9 + 21div	✓	In vivo	SC

**Table 2 ijms-22-02173-t002:** Overview of experimental groups for microinjection. PC of age groups 2 and 3 were assigned to all experimental setups A–D. All injections were performed simultaneously with pEGFP-C1.

Setup	Injection	Incubation (48 h)
A	pLifeAct-Tag-RFP	untreated
B	pLifeAct-Tag-RFP	VEGF
C	pLKO-miR204-DsRed	untreated
D	pLKO-miR204-DsRed	VEGF

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
