# Peer review of "Disabling VEGF-Response of Purkinje Cells by Downregulation of KDR via miRNA-204-5p"

_ijms, 2021, doi:10.3390/ijms22042173_

Round 1

Reviewer 1 Report

Summary / significance: This article by Gehmayr et al. connects to previous work of the authors that study roles of miR in the context of brain development. This study focuses on the action of miR204-5p supposed to be capable to regulate VEGFR-2 expression in juvenile Purkinje cells, and to influence VEGF-mediated signalling activity in developing cerebellum. Functioning of miRs that specifically modulate growth factor action in the brain is of high interest for treating neuronal diseases. These agonists hold promise for treatment of patients by dampening/reactivating neuronal growth.

Experiments, data and conclusions: This is a detailed PC morphology and miR-containing vector transfection/microinjection study evaluating the impact and effect of miR204-5p in vitro. The authors use miR204-5p to test VEGFR-2 downregulation effect and VEGF signaling action reduction in rat cerebella in vitro. After confirmation of miR204-5p expression, the authors propose that miR204-5p downregulates VEGFR-2 expression and hence VEGF sensitivity in juvenile PC and, moreover, leading to blocked dendrite growth in more mature PC.

Level of interest/merit: The identification of novel powerful modulators of growth factor action in the brain and, hence, for use in medical applications is of high interest to the scientific readership. The experimental approaches shown are in principle well described and executed highly accurately, and the results drawn mainly from in vitro tests in rats seem to prove the action of miR204-5p finally blocking VEGF action in juvenile, but surprisingly also mature Purkinje cells. The authors conclude from the data that miR204-5p acts as a specific dendrite growth reducing agent.

English level: The English phrasing needs thorough improvements. Please check it with a spelling program or have a native speaker read the manuscript.

There are some concerns: Some sections and especially the Figures of the current manuscript have to be clearly improved to ameliorate the contents. Introduction and Discussion are rather lengthy and could be shortened, and the results could be described more stringently. A major weakness is the proof of real VEGR-2 amounts in the slice cultures. Several points in the manuscript should be addressed in order to make this paper publishable, and then it can be accepted with essential revisions.

Major comments:

Results Fig.1b: why didn't the authors do a Western blot after the elaborate laser microdissection to check the VEGFR protein levels?

Results Fig.2 and 3: Regarding IF images, these presentations should be improved. The stained antigens (calbindin, VEGFR2, DAPI) should be shown in the image, while the other inscriptions BCL, PCL, MCL are rather distracting. Reduced VEGFR-2 expression shown in green at p15 or p30 is not clear: there is a strong green signal visible in layer2. The fine arrows indicating the Bergmann fibers are hardly visible. The same applies to Figure 3: VEGFR-2 downregulation is not shown convincingly. Fig.3e is very blurry, do you have a better image? the use of different image sizes (sometimes 50µm, sometimes 20µm) makes the morphological comparison difficult for the reader.

In line 205, the authors state that filamentous signals were detected, “but they cannot be clearly assigned to Bergmann glia cells in that case”. This is stated without explanation. Please clarify.

Results Fig.4, line 228f: VEGFR-2 expression should be also analyzed in Western blotting to show miR204-5p effectiveness on protein level. The conclusion is drawn in line230, but should be put at the end of the section. Regarding statistic significances shown in Fig. 4b and c, why did these expression differences not show a higher significance? Especially when compared with the 3-star significances indicated in Fig.5c and f. Please clarify.

Results Fig.5: similar to Fig. 2 and 3, Mainly, Figure panel arrangement and inscriptions are confusing: What do the pLifeActTag RFP and pEGFP-C1 injections visualize? Which panels show the miR204-injected dendrites, and which the VEGF-treated ones? What were the VEGF concentrations used in the stimulations? Inscriptions are far too small and hardly readable. Please enlarge.

Results Fig.5 described on page 8/9: The readout of the experiment is only morphometric. Although the authors claim an effect of VEGFR-2 action in juvenile PC, a major effect of dendrite reduction is seen rather in mature PC. How can that be explained in light of the previous data showing enhanced VEGFR-2 expression in juvenile PC? There is speculation about VEGFR-2 levels in the discussion (line 402f), but the only clear indication would be the performance of a Western blot experiment, also to show effectiveness of miR204-5p in downregulating VEGFR-2. Results in sections 2.4.2. and 2.4.3. are listed without comment, introductory, and concluding sentences are missing. Instead, there is a lengthy discussion.

Minor comments:

1) The title could be more captivating. VEGFR2 ist the more common name.

2) Introduction: Lines 66-72 can be omitted, are more like a review, the text is too detailed and does not relate to the actual study. Line 100f: the work from Pieczora et al. is essential and should be highlighted as initiative of the study. Sentences lines 109 and 111 should be swapped in order, which makes more sense.

Author Response

Dear Reviewer,

Thank you for your letter and please find enclosed the revised version of our manuscript, which we would like to publish in IJMS.

I would like to thank you and the editor on behalf of all co-authors for the critical reading of our manuscript. We are confident after improving the manuscript that we can entirely resolve the doubts of the reviewer.

We thank you for your effort and all your helpful critical advices which we tried to set up the best way and which help to improve the manuscript.

In the following I will address all the specific remarks point by point.

Yours sincerely,

Carsten Theiss

Reviewer 2 Report

Gehmeyr et al. present an interesting work “Disabling VEGF-response of Purkinje cells by downregulation of KDR via miRNA-204-5p” on the negative regulation of the VEGF-response of Purkinje cells by the miRNA-204. They focus on the VEGFR-2 receptor and have developed nice organotypic cultures to show the effects of the sh-miR204 at different stages of PC development. The presented data is clear but some controls are lacking before further acceptance of this article. The main results are:

  • miR204-5p has a negative effect on VEGF sensitivity
  • there is a decrease of dendritic growth upon treatment with this miR in juvenile PC
  • a shrinkage of dendrites upon miR204 treatment in mature PCs is observed despite VEGF treatment

The authors should answer some key issues concerning the results they present.

Major points

  • Lack of proper controls for the miR204-5p. The empty vector is not sufficient to show the specificity of this effect. The authors should transfect a mutated version of the miR204 sequence in the same vector and should that in this case there is no effect. A shuffled sequence could be used for this.
  • The authors claim that the effect of the miR204 is on the VEGFR2 and that this explains the different observed phenotypes. They use a N2A cell line and show that the mRNA levels of KDR decrease upon sh-miR204 overexpression. No such evidence is shown for the organotypic cultures (Figure 5). Here the effect on dendritic growth is quite mild and no direct evidence of a decrease in KDR levels is shown. Thereby, the observed effects could be due to other factors than KDR as there are other known targets for this miR204. The authors should provide further evidence of this link with VEGF:
    • An shRNA downregulating KDR could be a good control to monitor the effect of the decrease of this factor on dendritic growth.
    • Levels of KDR should be monitored in the cultured cells
    • More cells should be counted as the effects are very mild (9 cells is very limiting for the statistical values).

Minor points:

- Figure1: 2 different graphs with adapted scales will better demonstrate the data and the p values. A graph for the KDR (scaled at 1.2) and another for the other factors (scaled at 0.1 max) should be made.

- l.231: The authors overinterpret their data. Indeed, this experiment does not demonstrate the “interaction” but rather the “relationship” between miR204 and VEGFR2 as no direct binding assays have been carried out in this work.

- English language: The manuscript should be thoroughly re-read for syntaxe errors and phrasing (l.41 belong to, l.43 verb lacking, l.67 and l.111 sentences not clear, l.216 proof/prove? etc….)

Author Response

Dear Reviewer 2,

Thank you for your letter and please find enclosed the revised version of our manuscript, which we would like to publish in IJMS.

I would like to thank you and the reviewers on behalf of all co-authors for the critical reading of our manuscript. We are confident after improving the manuscript that we can entirely resolve the doubts of the reviewers.

We thank you for your effort and all your helpful critical advices which we tried to set up the best way and which help to improve the manuscript.

In the following I will address all the specific remarks point by point.

Yours sincerely,

Carsten Theiss

Round 2

Reviewer 1 Report

The revised manuscript by Gehmayr et al., addresses essentially the comments and concerns. The authors have elaborated points of criticism in a detailed manner. They have rearranged and renamed the sections of the text in such a way that it is now more clear to read, and they have improved graphic representations and inscriptions.

Author Response

Dear Reviewer,

We would like to thank you very much for the very valuable suggestions and comments. We are pleased that you now accept the manuscript for publication.

Yours sincerely

Carsten Theiss

Reviewer 2 Report

The authors have made the effort to answer point-by-point the different concerns. They provide some valuable arguments that reinforce their findings and can be taken into account considering the circumstances. The authors have also modified their conclusions to avoid over-interpretation of the findings.

Taking into consideration the COVID crisis and the linked difficulties in providing extra experimental data, the manuscript can be considered as sound for publication. Some controls would have been nice to reinforce the findings.

Some minor points should be checked :

- the p value is missing for KDR levels in Fig 1b

- Figure2 : please check scale bars for accuracy. Is it truely 1000microm 

Some language errors are still there :

- Line 16. « processes have …. »

- line 31 : the results of « this » study

- Line 179 « immeasurable » is a missense. Does the author mean cannot be measured ? « undetectable »  ?

Author Response

Dear Reviewer,

We would like to thank you very much for the very valuable suggestions and comments. We are very pleased that the revised manuscript is considered worthy of publication in IJMS, except for minor changes.

In the following I will address all the specific remarks point by point:

The authors have made the effort to answer point-by-point the different concerns. They provide some valuable arguments that reinforce their findings and can be taken into account considering the circumstances. The authors have also modified their conclusions to avoid over-interpretation of the findings. Taking into consideration the COVID crisis and the linked difficulties in providing extra experimental data, the manuscript can be considered as sound for publication. Some controls would have been nice to reinforce the findings.

We also thank you very much for this comment and are pleased that you now accept the revised manuscript for publication.

Some minor points should be checked :

- the p value is missing for KDR levels in Fig 1b

Thank you very much for this advice. The p-value (p ≤0.0001) in Figure 1b has now been entered.

- Figure2 : please check scale bars for accuracy. Is it truely 1000microm 

Thank you very much for this as well. It is 100 µm (and not 1000 µm). The error has been corrected in the figure legend.

Some language errors are still there :

- Line 16. « processes have …. »

- line 31 : the results of « this » study

- Line 179 « immeasurable » is a missense. Does the author mean cannot be measured ? « undetectable »  ?

Thank you very much. The linguistic errors have been corrected.

Yours sincerely

Carsten Theiss